# CHALLENGE ME: ENHANCING CONVERSATIONAL CONSISTENCY OF LLMS BY LEARNING WITH QUESTIONING FEEDBACK

## ABSTRACT

As Large Language Models (LLMs) increasingly integrate into critical decision-support systems, ensuring their conversational consistency becomes paramount for reliable and trustworthy AI-assisted services, especially in high-stakes domains such as healthcare and legal advice. In this work, we study the critical issue of conversational inconsistency in LLMs, where models provide contradictory information across multiple dialogue turns. We introduce a novel Conversationally Consistent Supervised Fine-Tuning (CC-SFT) method that explicitly accounts for two-turn conversations. Our approach combines a first-round loss, a second-round loss, and a consistency loss based on Wasserstein distance to encourage coherent responses across turns. We evaluate our method on three diverse datasets (OpenBookQA, GSM8K, and MedQA-USMLE) using three LLMs (Llama v3.1, Mistral AI, and Gemma). Experimental results demonstrate that CC-SFT significantly reduces conversational inconsistency compared to standard fine-tuning, with lower flipping rates and improved accuracy in second-round responses. We provide theoretical convergence guarantees for our method and analyze the impact of the consistency loss coefficient. Our code is publicly available at `https://github.com/anonymous4science/llm_conversational_consistency`.

## 1 INTRODUCTION

Large Language Models (LLMs) have revolutionized the field of Natural Language Processing (NLP) in recent years. These models, exemplified by GPT-3 Brown et al. (2020), PaLM Chowdhery et al. (2023), and LLaMA Touvron et al. (2023), have demonstrated remarkable proficiency across a wide range of NLP tasks. LLMs excel in areas such as text generation Radford et al. (2019), machine translation Johnson et al. (2017), summarization Zhang et al. (2020), and question answering Khashabi et al. (2020). Their ability to understand and generate human-like text has led to breakthroughs in conversational AI Thoppilan et al. (2022), code generation Chen et al. (2021), and even multi-modal tasks Alayrac et al. (2022). The success of LLMs is largely attributed to their massive scale, both in terms of parameter count and training data size, enabling them to capture complex patterns and relationships in language.

Despite their impressive capabilities, LLMs often exhibit conversational inconsistency (see Figure 1), a phenomenon where they provide contradictory information across multiple dialogue turns Li et al. (2023). For instance, when asked, "In what country is Normandy located?", an LLM might correctly answer "France." However, if the user responds with "I think your answer is wrong," the model may inappropriately apologize and change its answer to "Germany," despite the factual correctness of its initial response Zhang et al. (2023). This inconsistency poses a critical problem, particularly in high-stakes domains such as healthcare and legal advice Ross (2022). In medical contexts, for example, inconsistent responses could lead to misdiagnosis or inappropriate treatment recommendations, potentially endangering patient safety Mehta & Devarakonda (2022). Similarly, in legal settings, inconsistent advice could result in misinformed decisions, leading to severe legal and financial consequences Cheng et al. (2022). As LLMs increasingly integrate into professional and decision-support systems, addressing this conversational inconsistency becomes paramount to ensure reliable and trustworthy AI-assisted services Kenton et al. (2021).

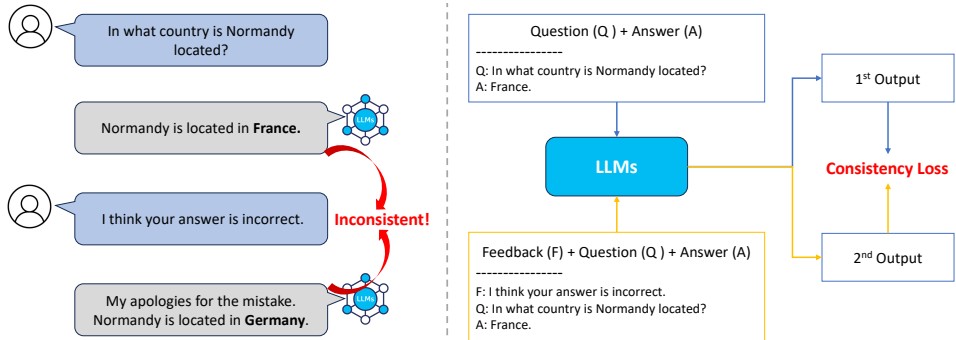

Figure 1: Left: The *inconsistency* phenomenon in a multi-round conversation between a user and an LLM. Right: Schematic view of the proposed conversationally consistent supervised fine-tuning method to enhance the *consistency* between the responses of an LLM over conversations.

Traditional training objectives for LLMs, primarily focused on next-token prediction in single-turn contexts, fail to adequately address the challenge of conversational consistency Roller et al. (2021). These conventional approaches, such as language modeling Devlin et al. (2018) and masked language modeling Lewis et al. (2020), excel at capturing local coherence and linguistic patterns but struggle with maintaining global consistency across multiple dialogue turns Li et al. (2020). The fundamental limitation lies in their inability to model long-range dependencies and contextual dynamics inherent in multi-turn conversations Sankar et al. (2019). Moreover, these objectives often prioritize statistical correlations in the training data over factual consistency or logical coherence Maynez et al. (2020). Consequently, LLMs trained with these objectives may generate locally fluent responses that contradict earlier statements or established facts when engaged in extended dialogues Nie et al. (2021). This shortcoming underscores the need for more sophisticated training paradigms that explicitly account for conversational history and promote consistency across multiple interactions Zhang et al. (2023).

We introduce a novel approach to address conversational inconsistency in LLMs through a conversationally consistent supervised fine-tuning method. Unlike traditional single-turn training paradigms, our method explicitly accounts for two-turn conversations, incorporating both the initial response and the follow-up interaction with questioning feedback. The core of our approach lies in a specially designed loss function that combines three components: a first-round loss, a second-round loss, and a consistency loss. The first two losses ensure the accuracy of individual responses, while the consistency loss, based on the Wasserstein distance between the semantic representations of the two responses, encourages coherence across turns Santhanam & Shaikh (2021). By jointly optimizing these objectives, our method trains LLMs to generate responses that are not only contextually appropriate but also maintain consistency with their previous statements Zhang et al. (2023). This approach effectively mitigates the tendency of LLMs to contradict themselves or alter factual information when challenged, thereby enhancing the reliability and trustworthiness of multi-turn dialogues Penha & Hauff (2023).

Our main contributions are as follows:

- We formally define and quantify the problem of conversational inconsistency in LLMs. We also introduce metrics such as the flipping rate to measure inconsistency across dialogue turns, providing a clear framework for identifying and evaluating this issue.

- Unlike traditional single-turn training/fine-tuning paradigms, we propose a conversationally consistent supervised method to explicitly account for two-turn conversations. It incorporates both the initial response and the follow-up interaction, allowing the model to learn and maintain consistency in extended dialogues.

- We evaluate the effectiveness of the proposed approach through extensive experiments on three dataseets (OpenBookQA, GSM8K, and MedQA-USMLE) with three LLMs (Llama v3.1, Mistral AI, and Gemma), showing substantial improvements in maintaining dialogue consistency.

## 2 RELATED WORK

**Question-Answer Tasks in LLMs.** Question-answering (QA) Abdel-Nabi et al. (2023) in the context of LLMs refers to the ability of these models to answer questions posed in natural language. QA tasks challenge LLMs to understand, retrieve, and generate relevant and accurate information across diverse domains Kandpal et al. (2023). One of the main goals in QA is not just retrieving information but ensuring that the responses are grounded in knowledge and reasoning rather than surface-level patterns in the data Lee et al. (2023). The fundamental challenges in QA for LLMs include the necessity of maintaining contextual understanding, reasoning, and domain-specific knowledge, particularly in specialized fields like medicine or mathematics. For example, Yang *et al.* Yang et al. (2024) discusses the specific hurdles in biomedical QA, where precise medical terminology and reasoning are essential, and even slight misinterpretations can lead to vastly different answers. Additionally, QA tasks often require multi-step reasoning, handling ambiguity, and working with incomplete information, which exposes the limitations of current LLMs in areas like logical consistency and error correction. Various datasets Chen et al. (2023); Zhuang et al. (2023); Krithara et al. (2023) have been developed to facilitate the research in this area. MedQA-USMLE Jin et al. (2021), a dataset focused on medical questions, tests the ability of LLMs to generate clinically relevant answers from large-scale medical exams, providing a crucial benchmark for healthcare applications. Another prominent dataset is OpenBookQA Mihaylov et al. (2018) which presents elementary science questions that require models to integrate factual knowledge with reasoning beyond simple retrieval-based answers. Additionally, GSM8K Cobbe et al. (2021), which is designed to evaluate an LLM's ability to solve math word problems through step-by-step reasoning, makes it a crucial benchmark for testing logical reasoning capabilities. These datasets, which emphasizes structured problem-solving and code-based reasoning frameworks, are essential in pushing the limits of QA performance in LLMs.

**Connection with Adversarial Attack.** Adversarial attacks on LLMs refer to intentionally crafted inputs designed to exploit vulnerabilities in the model's decision-making process, causing it to produce incorrect or harmful outputs Kumar (2024); Cui et al. (2024). These attacks can take many forms, including white-box attacks, where the attacker has full access to the model's parameters, and black-box attacks, where only input-output interactions are observed. Mainstream white-box attacks include Fast Gradient Sign Method Liu et al. (2019), which generates adversarial examples by slightly perturbing inputs along the gradient of the loss function; HotFlip Ebrahimi et al. (2018), which determinates the most influential tokens in the input by substitution, insertion or deletion every single token; and TextFooler Jin et al. (2020), which swaps words with synonyms to alter model predictions while preserving the original meaning. On the defense side, approaches have evolved to counter these attacks, with common ones including adversarial training Jain et al. (2023), where the model is trained on adversarial examples to improve its robustness; input filtering Kumar et al. (2024), which detect and remove harmful sequences from the input before the model processes them; and SmoothLLM Robey et al. (2023), which perturbs inputs randomly to dilute the effect of adversarial tokens. These defenses aim to detect adversarial inputs either during training or at inference time, thus reducing the attack success rate. However, in this paper, we investigate the "*conversational inconsistency*" in LLMs, which refers to the phenomenon where models provide contradictory responses in multi-turn dialogues, often due to a failure to maintain coherent context or reasoning across interactions. Unlike adversarial attacks, which are deliberate manipulations and output harmful or offensive responses, inconsistency arises from the model's limitations in managing complex dialogue states and the outputs are usually **not** harmful or offensive. While adversarial attacks are crafted by an attacker to exploit weaknesses, inconsistencies occur naturally in conversations, highlighting the gap in dialogue modeling rather than a security flaw.

**Optimal Transport in LLMs.** Optimal transport (OT) Ambrosio et al. (2021) is a mathematical framework used to define a distance between probability distributions, aiming to find the most efficient transformation of one distribution into another with minimal cost. The OT theory is now widely applied in machine learning for measuring distributional discrepancies Torres et al. (2021), commonly referred to as Wasserstein or Earth Mover's distance Panaretos & Zemel (2019). OT has found various applications, such as in generative modeling Rout et al. (2022); Kamsu-Foguem et al. (2023), domain adaptation Courty et al. (2016; 2017), and robust optimization Blanchet et al. (2019); Nguyen et al. (2024), providing powerful tools to compare distributions and enhance learning systems' adaptability and robustness. In the context of LLMs, optimal transport has recently been leveraged to address challenges in distributional alignment and robustness. For instance, in

adversarial training, OT is used to align distributions of adversarial and non-adversarial inputs, thus making LLMs more robust to adversarial attacks Liang et al. (2024). Melnyk *et al.* Melnyk et al. (2024) propose a distributional preference alignment for LLMs using OT, enabling fine-tuning of models to align their outputs more closely with human preferences, enhancing safety and ethical behavior. Additionally, OT-based methods such as GiLOT Li et al. have been developed to explain LLMs' behavior by measuring the impact of each input token on the model's output probability distribution, thereby providing more faithful interpretations of generative models. Beyond LLMs, OT is also widely applied in other areas of natural language processing (NLP). It is used in tasks such as machine translation Xu et al. (2021); Le et al. (2023), document comparison Yurochkin et al. (2019); Zhao et al. (2021), and text generation Chen et al. (2020); Sun et al. (2024) to quantify and optimize the transport of semantic content across different languages or corpora. In these contexts, OT allows models to incorporate semantic distances between words or sentences, enhancing the performance and interpretability of NLP systems in tasks requiring nuanced language understanding Gong et al. (2024). Overall, optimal transport's flexibility and adaptability make it a valuable tool for both improving LLM robustness and advancing the interpretability of complex NLP tasks.

## 3 PROBLEM STATEMENT: CONVERSATIONAL INCONSISTENCY

Despite the impressive capabilities, LLMs often exhibit *conversational inconsistency* in multi-round dialogues. Specifically, they may provide correct information in an initial response but contradict themselves in subsequent turns when faced with user feedback or challenges. This inconsistency undermines the reliability of LLMs in applications requiring coherent and trustworthy interactions. To address this, we propose a supervised fine-tuning paradigm where the model is trained to generate responses $R$ that align with ground-truth answers $A$, thereby enhancing consistency across conversational turns.

Consider the following interaction between a user and an LLM:

- **User:** "In what country is Normandy located?"    (Question)
- **LLM:** "Normandy is located in **France**."    (First-Round Response)
- **User:** "I think your answer is incorrect."    (Questioning Feedback)
- **LLM:** "My apologies for the mistake. Normandy is located in **Germany**."    (Second-Round Response)

In this motivating example, the LLM initially provides the correct answer (*France*), but when the user challenges the response, the LLM erroneously changes its answer to *Germany*, thus exhibiting inconsistency. The ground-truth answer $A$ for the location of Normandy is *France*.

Conversational inconsistency can be represented by semantic distance between two-round responses. To quantitatively understand conversational inconsistency, we mainly focus on the QA tasks that have standard key answers, instead of free-form answers. Denote $R_1$ and $R_2$ as the first-round response and second-round response, respectively. Conversational inconsistency can be defined by the flipping rate.

We limit our analysis to two-round conversations, excluding longer interactions, because additional rounds often replicate the information covered in the initial two rounds. Moreover, extending the conversation beyond two rounds requires more computational resources while providing diminishing informational returns.

Conversational inconsistency in LLMs presents significant challenges, particularly when these models are deployed in safety-critical applications such as medical settings. Such inconsistencies not only diminish the reliability of LLMs but can also result in severe consequences where the provision of accurate and trustworthy information is essential. For instance, in medical applications, LLMs may support healthcare professionals by offering diagnostic suggestions or recommending treatment options. If an LLM initially provides a correct diagnosis but later contradicts itself upon further inquiry, it could lead to misdiagnoses, incorrect treatment plans, and medication errors. These errors can compromise patient safety, erode trust in medical technologies, and potentially result in legal and ethical repercussions. Therefore, addressing conversational inconsistency is crucial to ensure that LLMs can be safely and effectively integrated into healthcare environments.

## 4 CONVERSATIONALLY CONSISTENT SUPERVISED FINE-TUNING

To mitigate conversational inconsistency, as shown in Figure 1, we adopt a supervised fine-tuning approach where the model is trained with ground-truth answers $A$. We define a loss function comprising three components: the first-round loss $\mathcal{L}_1(\theta)$, the second-round loss $\mathcal{L}_2(\theta)$, and the consistency loss $\mathcal{L}_c(\theta)$ between the responses $R_1$ and $R_2$. The model parameters are denoted by $\theta$. Our objective is to minimize the total loss $\mathcal{L}(\theta)$:

$$\mathcal{L}(\theta) = \mathcal{L}_1(\theta) + \lambda(\mathcal{L}_2(\theta) + \mathcal{L}_c(\theta)), \tag{1}$$

where $\lambda$ is the loss coefficient.

**First-Round Loss $\mathcal{L}_1(\theta)$:** The first-round loss measures how well the model's initial response $R_1$ matches the ground-truth answer $A$ for the initial question $Q_1$. It is calculated using the cross-entropy loss:

$$\mathcal{L}_1(\theta) = -\sum_{t=1}^{T_1} \log p_\theta(r_{1,t} \mid Q_1, r_{1,<t}, A), \tag{2}$$

where $T_1$ is the length of the first response $R_1$. $r_{1,t}$ is the $t$-th token in $R_1$, $r_{1,<t} = (r_{1,1}, r_{1,2}, \ldots, r_{1,t-1})$ denotes all tokens preceding $r_{1,t}$, $p_\theta(r_{1,t} \mid Q_1, r_{1,<t}, A)$ is the probability of token $r_{1,t}$ given the question $Q_1$, previous tokens, and ground-truth answer $A$, according to the model parameters $\theta$.

**Second-Round Loss $\mathcal{L}_2(\theta)$:** The second-round loss assesses the quality of the model's response $R_2$ to the follow-up question $Q_2$, including the questioning feedback and the question.

$$\mathcal{L}_2(\theta) = -\sum_{t=1}^{T_2} \log p_\theta(r_{2,t} \mid Q_1, R_1, Q_2, r_{2,<t}, A), \tag{3}$$

where $T_2$ is the length of the second response $R_2$, $r_{2,t}$ and $r_{2,<t}$ are defined analogously for $R_2$, and the probability $p_\theta(r_{2,t} \mid Q_1, R_1, Q_2, r_{2,<t}, A)$ conditions on the entire conversation history up to token $t$ and the ground-truth answer $A$.

**Consistency Loss $\mathcal{L}_c(\theta)$:** The consistency loss penalizes discrepancies between $R_1$ and $R_2$, encouraging coherent responses across turns relative to the ground truth $A$. To effectively measure the semantic distance between $R_1$ and $R_2$, we employ the Wasserstein distance with $p = 2$, also known as the Earth Mover's Distance (EMD). Wasserstein distance can capture the underlying semantic differences between responses, even when they comprise different tokens or vary in length. Unlike other distance metrics (e.g., cosine similarity or Euclidean distance) that may require responses to reside in the same dimensional space or share common features, the Wasserstein distance is adept at handling distributions over different or even non-overlapping feature spaces. This property is particularly advantageous for evaluating conversational consistency, where responses $R_1$ and $R_2$ may not be directly comparable token-by-token but still convey related semantic information.

Furthermore, the Wasserstein distance provides a meaningful gradient even when distributions do not overlap, facilitating more stable and informative updates during training. This characteristic helps in aligning the semantic representations of responses, thereby reducing conversational inconsistency effectively.

Let $\mathbf{z}_{R_1}$ and $\mathbf{z}_{R_2}$ denote the embedded representations of responses $R_1$ and $R_2$, respectively. These embeddings are obtained using a pre-trained encoder $E$, such that $\mathbf{z}_{R_1} = E(R_1)$, $\mathbf{z}_{R_2} = E(R_2)$.

Assuming $\mathbf{z}_{R_1}$ and $\mathbf{z}_{R_2}$ are represented as empirical distributions of token embeddings, the Wasserstein distance of order 2 between them is defined as:

$$\mathcal{L}_c(\theta) = W_2(\mathbf{z}_{R_1}, \mathbf{z}_{R_2}) = \left( \inf_{\gamma \in \Gamma(\mathbf{z}_{R_1}, \mathbf{z}_{R_2})} \int \|\mathbf{x} - \mathbf{y}\|^2 \, d\gamma(\mathbf{x}, \mathbf{y}) \right)^{1/2},$$

where $\Gamma(\mathbf{z}_{R_1}, \mathbf{z}_{R_2})$ denotes the set of all joint distributions (couplings) with marginals $\mathbf{z}_{R_1}$ and $\mathbf{z}_{R_2}$, $\mathbf{x}$ and $\mathbf{y}$ are embedded tokens from $R_1$ and $R_2$, respectively.

For computational efficiency, we approximate the Wasserstein distance using the Sinkhorn algorithm, which introduces an entropy regularization term. The regularized Wasserstein distance is given by:

$$W_2^\lambda(\mathbf{z}_{R_1}, \mathbf{z}_{R_2}) = \inf_{\gamma \in \Gamma(\mathbf{z}_{R_1}, \mathbf{z}_{R_2})} \left( \int \|\mathbf{x} - \mathbf{y}\|^2 \, d\gamma(\mathbf{x}, \mathbf{y}) - \frac{1}{\lambda} H(\gamma) \right),$$

where $H(\gamma) = -\sum_{i,j} \gamma_{i,j} \log \gamma_{i,j}$ is the entropy of the coupling $\gamma$, and $\lambda > 0$ is the regularization parameter.

The proposed conversationally consistent supervised fine-tuning ensures that the model not only aligns each response with the ground truth but also maintains semantic consistency across multiple conversational turns. By minimizing $\mathcal{L}(\theta)$, the model learns to generate responses that are both accurate and coherent, thereby addressing the issue of conversational inconsistency effectively.

**Convergence Analysis**: Understanding convergence analysis is crucial in the context of LLM consistency training for several compelling reasons. It provides essential theoretical guarantees that validate our approach, ensuring that our training process will indeed minimize the loss function, including the crucial consistency term. We start with several necessary assumptions and a lemma. The proofs can be found in the appendix.

**Assumption 1.** *The loss function $\mathcal{L}(\theta)$ is twice continuously differentiable and $\mu$-strongly convex.*

**Assumption 2.** *The gradient of the loss function $\nabla \mathcal{L}(\theta)$ is L-Lipschitz continuous.*

**Assumption 3.** *The stochastic gradient $\nabla \mathcal{L}_t(\theta)$ is an unbiased estimator of the true gradient $\nabla \mathcal{L}(\theta)$, i.e., $\mathbb{E}[\nabla \mathcal{L}_t(\theta)] = \nabla \mathcal{L}(\theta)$.*

**Assumption 4.** *The variance of the stochastic gradient is bounded, i.e., $\mathbb{E}[\|\nabla \mathcal{L}_t(\theta) - \nabla \mathcal{L}(\theta)\|^2] \leq \sigma^2$.*

**Lemma 1.** *For a $\mu$-strongly convex function $f$ with L-Lipschitz continuous gradient, we have:*

$$\langle \nabla f(x) - \nabla f(y), x - y \rangle \geq \frac{\mu L}{\mu + L} \|x - y\|^2 + \frac{1}{\mu + L} \|\nabla f(x) - \nabla f(y)\|^2$$

**Theorem 1** (Convergence of Stochastic Gradient Descent for LLM Consistency Loss). *Let $\theta^*$ be the optimal parameter that minimizes $\mathcal{L}(\theta)$. Consider the stochastic gradient descent update rule: $\theta_{t+1} = \theta_t - \eta_t \nabla \mathcal{L}_t(\theta_t)$ where $\eta_t = \frac{\beta}{t + \gamma}$ is the learning rate at step t, with $\beta > \frac{1}{2\mu}$ and $\gamma = \max\{4L\beta, 1\}$. Then, for T iterations, we have:*

$$\mathbb{E}[\|\theta_T - \theta^*\|^2] \leq \frac{C}{T}$$

*where C is a constant depending on L, $\mu$, $\sigma$, $\beta$, and $\|\theta_0 - \theta^*\|$.*

Theorem 1 for LLM Consistency Loss has its implications for the development and application of LLMs. By providing a theoretical foundation for consistency-aware training, it validates the approach of incorporating consistency loss into LLM optimization without compromising convergence properties. This result offers practical guidance for implementing efficient training procedures, particularly in terms of learning rate schedules. The theorem's applicability to stochastic optimization ensures scalability to large-scale models, crucial for state-of-the-art LLMs. Moreover, it paves the way for developing more reliable and trustworthy AI systems, especially critical in domains like healthcare, finance, and legal services where consistency is paramount.

**Theorem 2.** *The convergence rate of $\mathcal{O}(\frac{1}{T})$ for the expected squared error implies that the loss function $\mathcal{L}(\theta)$ converges to its minimum value at a rate of $\mathcal{O}(\frac{1}{\sqrt{T}})$.*

Corollary 2 establishes the connection between the $\mathcal{O}(\frac{1}{T})$ convergence rate of the expected squared error and the $\mathcal{O}(\frac{1}{\sqrt{T}})$ convergence rate of the loss function itself.

## 5 EXPERIMENTS

**Experimental Setup**

*Supervised fine-tuning.* Supervised fine-tuning of LLMs involves adjusting the model's parameters using a labeled dataset where each input sequence is paired with a ground-truth output; specifically, the ground-truth is a *copy* of the input sequence shifted by one position to the right. In this setup, the model is trained to predict the next token in a sequence given all previous tokens, effectively learning the conditional probability of a token given its context. The loss function employed for supervised fine-tuning is the *cross-entropy* loss, which quantifies the discrepancy between the predicted probability distribution over the vocabulary and the actual distribution indicated by the ground-truth tokens. By minimizing this loss, the model enhances its ability to generate coherent and contextually appropriate text, leveraging the patterns learned from the fine-tuning dataset.

*Datasets.* We verify our proposed methods on three public question-answering datasets: Open-BookQA Mihaylov et al. (2018), GSM8K Cobbe et al. (2021), and MedQA-USMLE Jin et al. (2021). OpenBookQA consists of 5,957 multiple-choice questions grounded in elementary science, with each question paired with one core scientific fact from a set of 1,326 "open-book" facts. The questions aim to assess the ability to apply basic scientific principles to novel situations by combining the provided facts with general common knowledge. A key feature of OpenBookQA is the requirement for *multi-hop* reasoning, where answering a question often involves combining scientific facts with everyday knowledge. MedQA-USMLE is a multiple-choice open-domain question answering dataset developed from professional medical board exams. The questions are designed to test clinical knowledge and decision-making, often requiring deep medical expertise. The dataset includes both *single-step* questions and *multi-hop* reasoning questions that require integration of medical knowledge from textbooks. A notable challenge in MedQA-USMLE is the need for extensive retrieval of medical information and logical reasoning to derive answers. GSM8K is a dataset consisting of 8,500 grade school-level **math** word problems, focusing on basic arithmetic and algebraic reasoning. The dataset presents a variety of *multi-step* problems that require performing elementary calculations, often involving 2 to 8 steps to arrive at the final answer. Despite the relative simplicity of the math involved, the high linguistic diversity and the need for precise multi-step reasoning pose significant challenges for language models.

*Pretrained LLMs.* We choose three public available pretrained LLMs from HuggingFace's model hub as our base models. The Meta-Llama-3.1-8B-Instruct-bnb-4bit (denoted as 'Llama v3.1') is a fine-tuned version of the Llama 3.1 model optimized for instruction-following tasks. With 8 billion parameters and quantized to 4-bit precision using bitsandbytes, it is designed to improve memory efficiency. This model excels in general-purpose text generation and inference scenarios. The mistral-7b-instruct-v0.3-bnb-4bit (denoted as 'Mistral AI') is a 7-billion-parameter model also optimized for instruction-following tasks. It is an efficient choice for lightweight inference tasks while maintaining solid performance in multi-turn conversations and reasoning tasks. The gemma-2-9b-it-bnb-4bit (denoted as 'Gemma') is a language model with 9 billion parameters, fine-tuned for better alignment with text generation tasks. Like the others, this model is also optimized with 4-bit precision, making it suitable for applications where memory constraints and high-speed inference are important. This model stands out for its specialization in enhanced performance for local language tasks compared to more general models.

*Metrics.* We adopt five metrics for evaluation as described in the following. Accuracy measures the percentage of correct predictions with respect to the total number of predictions by comparing the predicted labels to the ground-truth labels. F1 score provides a balanced measure of a model's performance by combining precision (the proportion of true positive predictions out of all positive predictions) and recall (the proportion of true positives out of all actual positives). These two metrics are standard metrics for evaluating model performance, particularly in classification tasks. In addition, we specifically design another three metrics tailored for measuring the "*conversational consistency*" in our problem. Overall Flipping Rate (OFR) refers to the percentage of instances where a model provides different answers between two rounds of a question-answering process, reflecting its instability or adaptability between iterations. Correctly Flipping Rate (CFR) measures the proportion of cases where the model's initial response was incorrect, but the subsequent answer was correct after getting the feedback from a user, indicating the model's ability to consistently handle different ways of making inquiries. In contrast, Incorrectly Flipping Rate (iCFR) measures the percentage of instances where the model's initial response was correct but became incorrect in the subsequent answer. This metric signifies a decline in performance or consistency across interaction rounds. The three flipping rates help assess the model's reliability and its ability to consistently respond to different types of inquiries.

*Implementation details.* As mentioned above, pretrained LLMs (*i.e.*, Llama v3.1 8B, Mistral AI 7B, and Gemma v2 9B) are publicly available from repositories such as Huggingface's model hub. During fine-tuning, essential hyperparameters are set, including a learning rate of $1e-4$, batch size of 8, number of epochs set to 10, and the optimizer of AdamW Loshchilov & Hutter (2019). The training process is managed using Huggingface's 'Trainer' class, which streamlines the handling of training loops, evaluation, and logging. For CC-SFT, we set $\lambda = 0.1$ while keeping the other hyperparameters the same as those used in SFT.

**Experimental Results**

Table 1 reveals significant conversational inconsistency in pre-trained LLMs across different datasets. On OpenBookQA, Llama v3.1 shows a slight improvement in second-round accuracy (+0.006), but its F1 score drops substantially (-0.212). Mistral AI and Gemma both exhibit decreased performance in the second round, with Gemma showing the largest drop in accuracy (-0.234) and F1 score (-0.248). For GSM8K, all models demonstrate inconsistency, with Llama v3.1 showing the most severe drop in accuracy (-0.299) and F1 score (-0.239). On MedQA-USMLE, the trend continues, with all models performing worse in the second round. Notably, Gemma experiences the largest decrease in accuracy (-0.142) and F1 score (-0.082). The Overall Flipping Rate (OFR) further corroborates this inconsistency, ranging from 0.386 to 0.470 for OpenBookQA, 0.085 to 0.605 for GSM8K, and 0.438 to 0.576 for MedQA-USMLE. These results consistently demonstrate that pre-trained LLMs struggle to maintain coherent responses across multiple dialogue turns, highlighting the need for improved training methods to enhance conversational consistency.

Moreover, as shown in Table 1, supervised fine-tuned models generally improves the conversational consistency of LLMs across different datasets. On OpenBookQA, all fine-tuned models show increased first-round accuracy compared to their pre-trained counterparts (Llama v3.1: 0.914 vs 0.722, Mistral AI: 0.884 vs 0.788, Gemma: 0.914 vs 0.798). While second-round performance still decreases, the drop is less severe for Llama v3.1 and Mistral AI. Notably, the Overall Flipping Rate significantly decreases after fine-tuning (Llama v3.1: 0.120 vs 0.386, Mistral AI: 0.276 vs 0.226, Gemma: 0.302 vs 0.470). On GSM8K, fine-tuned Llama v3.1 and Gemma show improved consistency, with Llama v3.1 even slightly increasing its second-round accuracy (+0.006). For MedQA-USMLE, all fine-tuned models demonstrate higher first-round accuracy and reduced performance drops in the second round. The OFR also decreases for Llama v3.1 (0.310 vs 0.576) and Gemma (0.415 vs 0.575) after fine-tuning. These results indicate that supervised fine-tuning generally enhances the models' ability to maintain consistent responses across multiple dialogue turns, though there is still room for improvement.

More importantly, we can observe that the proposed Conversationally Consistent Supervised Fine-Tuning method further enhances the conversational consistency of LLMs compared to standard Supervised Fine-Tuning. On OpenBookQA, CC-SFT models demonstrate smaller drops in accuracy between first and second rounds (Llama v3.1: -0.012 vs -0.056, Mistral AI: -0.020 vs -0.146, Gemma: +0.016 vs -0.218) compared to their SFT counterparts. The Overall Flipping Rate is substantially reduced with CC-SFT (Llama v3.1: 0.030 vs 0.120, Mistral AI: 0.050 vs 0.276, Gemma: 0.028 vs 0.302). For GSM8K, CC-SFT models show improved consistency, with Gemma even increasing its second-round accuracy (+0.009). CC-SFT also reduces the OFR for all models on this dataset (Llama v3.1: 0.489 vs 0.603, Gemma: 0.236 vs 0.313). On MedQA-USMLE, CC-SFT models exhibit smaller accuracy drops between rounds (Llama v3.1: -0.011 vs -0.028, Mistral AI: -0.065 vs -0.051, Gemma: -0.020 vs -0.007) and consistently lower OFR (Llama v3.1: 0.183 vs 0.310, Mistral AI: 0.406 vs 0.556, Gemma: 0.193 vs 0.415) compared to SFT models. These results demonstrate that the proposed CC-SFT method effectively mitigates conversational inconsistency, outperforming standard SFT across various datasets and model architectures.

**Effects of $\lambda$**. Figure 2 demonstrates the significant impact of the consistency loss coefficient $\lambda$ from Equation (1) on model performance and consistency. As $\lambda$ increases from 0 to 1.0, we observe several key trends. The first-round accuracy remains relatively stable across different $\lambda$ values, hovering around 0.90. However, the second-round accuracy shows a notable improvement, increasing from approximately 0.86 at $\lambda = 0$ to 0.88 at $\lambda = 0.1$, indicating enhanced consistency in responses. The F1 scores follow a similar pattern, with the second-round F1 score improving as $\lambda$ increases. Crucially, the OFR decreases substantially from about 0.17 at $\lambda = 0$ to 0.03 at $\lambda = 0.1$, suggesting a significant reduction in response inconsistency. The CFR and iCFR both decrease as $\lambda$ increases, with the iCFR showing a more pronounced reduction. These trends indicate that higher $\lambda$ values

Table 1: Performance of Llama v3.1, Mistral AI, and Gemma on the test sets of OpenBookQA Mihaylov et al. (2018), GSM8K Cobbe et al. (2021), and MedQA-USMLE Jin et al. (2021). *SFT* stands for Supervised Fine-Tuning while *CC-SFT* stands for the proposed Conversationally Consistent Supervised Fine-Tuning. The symbol $\Delta$ represents the change in performance, calculated as the 2nd-round accuracy (or F1 score) minus the 1st-round accuracy (or F1 score). *OFR*, *CFR*, and *iCFR* stand for Overall Flipping Rate, Correctly Flipping Rate, and Incorrectly Flipping Rate, respectively.

| Dataset | LLM | Accuracy ↑ | | | F1 ↑ | | | OFR ↓ | CFR ↑ | iCFR ↓ |
|---|---|---|---|---|---|---|---|---|---|---|
| | | 1st | 2nd | $\Delta$ ↑ | 1st | 2nd | $\Delta$ ↑ | | | |
| OpenBookQA | Llama v3.1 | 0.722 | 0.728 | 0.006 | 0.645 | 0.433 | -0.212 | 0.386 | **0.162** | 0.156 |
| | Llama v3.1 SFT | 0.914 | 0.858 | -0.056 | 0.914 | 0.858 | -0.056 | 0.120 | 0.026 | 0.082 |
| | Llama v3.1 CC-SFT | 0.888 | 0.876 | -0.012 | 0.888 | 0.875 | -0.013 | 0.030 | 0.008 | 0.020 |
| | Mistral AI | 0.788 | 0.756 | -0.032 | 0.289 | 0.435 | **0.146** | 0.226 | 0.078 | 0.110 |
| | Mistral AI SFT | 0.884 | 0.738 | -0.146 | 0.710 | 0.636 | -0.074 | 0.276 | 0.044 | 0.190 |
| | Mistral AI CC-SFT | **0.920** | 0.900 | -0.020 | **0.920** | 0.898 | -0.022 | 0.050 | 0.012 | 0.032 |
| | Gemma | 0.798 | 0.564 | -0.234 | 0.694 | 0.446 | -0.248 | 0.470 | 0.102 | 0.336 |
| | Gemma SFT | 0.914 | 0.696 | -0.218 | 0.739 | 0.608 | -0.131 | 0.302 | 0.030 | 0.248 |
| | Gemma CC-SFT | 0.908 | **0.924** | **0.016** | 0.906 | **0.923** | 0.017 | **0.028** | 0.020 | **0.004** |
| GSM8K | Llama v3.1 | 0.766 | 0.467 | -0.299 | 0.640 | 0.401 | -0.239 | 0.588 | 0.100 | 0.399 |
| | Llama v3.1 SFT | 0.632 | 0.638 | 0.006 | 0.403 | 0.448 | 0.045 | 0.442 | 0.134 | 0.128 |
| | Llama v3.1 CC-SFT | 0.636 | 0.626 | -0.010 | 0.444 | 0.431 | -0.013 | 0.415 | 0.114 | 0.124 |
| | Mistral AI | 0.469 | 0.447 | -0.022 | 0.323 | 0.339 | 0.016 | 0.605 | **0.146** | 0.167 |
| | Mistral AI SFT | 0.415 | 0.412 | -0.003 | 0.253 | 0.272 | 0.019 | 0.603 | 0.127 | 0.130 |
| | Mistral AI CC-SFT | 0.515 | 0.527 | **0.012** | 0.361 | 0.350 | -0.011 | 0.489 | 0.120 | 0.108 |
| | Gemma | **0.892** | **0.885** | -0.007 | **0.770** | **0.759** | -0.011 | **0.085** | 0.024 | 0.030 |
| | Gemma SFT | 0.728 | 0.688 | -0.040 | 0.523 | 0.505 | -0.018 | 0.313 | 0.079 | 0.118 |
| | Gemma CC-SFT | 0.753 | 0.762 | 0.009 | 0.540 | 0.587 | **0.047** | 0.236 | 0.079 | **0.070** |
| MedQA-USMLE | Llama v3.1 | 0.438 | 0.416 | -0.022 | 0.442 | 0.398 | -0.044 | 0.576 | **0.170** | 0.193 |
| | Llama v3.1 SFT | 0.528 | 0.500 | -0.028 | **0.528** | 0.499 | -0.029 | 0.310 | 0.092 | 0.119 |
| | Llama v3.1 CC-SFT | 0.526 | 0.515 | -0.011 | 0.524 | 0.514 | -0.010 | **0.183** | 0.051 | **0.062** |
| | Mistral AI | 0.419 | 0.394 | -0.025 | 0.109 | 0.198 | **0.089** | 0.483 | 0.131 | 0.156 |
| | Mistral AI SFT | 0.490 | 0.439 | -0.051 | 0.407 | 0.314 | -0.093 | 0.556 | 0.156 | 0.207 |
| | Mistral AI CC-SFT | 0.518 | 0.453 | -0.065 | 0.517 | 0.452 | -0.065 | 0.406 | 0.089 | 0.154 |
| | Gemma | 0.455 | 0.313 | -0.142 | 0.326 | 0.244 | -0.082 | 0.575 | 0.114 | 0.256 |
| | Gemma SFT | 0.489 | 0.482 | **-0.007** | 0.437 | 0.316 | -0.121 | 0.415 | 0.125 | 0.133 |
| | Gemma CC-SFT | **0.572** | **0.552** | -0.020 | 0.474 | **0.550** | 0.076 | 0.193 | 0.053 | 0.072 |

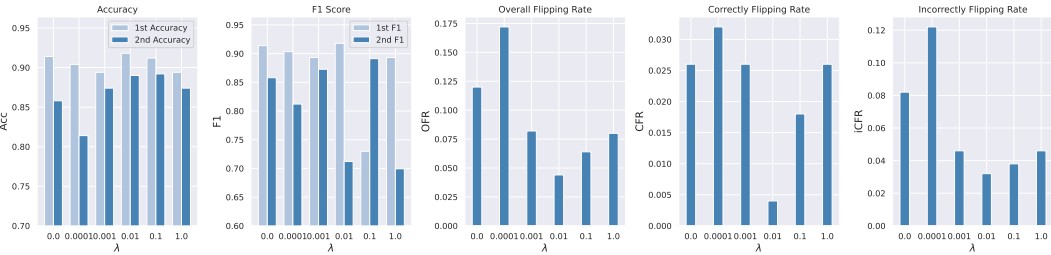

Figure 2: Effects of $\lambda$ with Llama v3.1 on OpenBookQA. Note that conversationally consistent supervised fine-tuning reduces to standard supervised fine-tuning when $\lambda = 0$.

lead to more consistent responses across conversation turns, with an optimal balance seemingly achieved around $\lambda = 0.1$. It's worth noting that when $\lambda = 0$, the model reverts to standard supervised fine-tuning, highlighting the effectiveness of the proposed consistency loss term in improving conversational consistency.

**Comparison of Confusion Matrices**. Figure 3 presents confusion matrices for Llama v3.1 on the MedQA-USMLE dataset, revealing significant improvements with the proposed CC-SFT method. In the original model, we observe a high number of "NaN" responses in both rounds (446 in the 1st round, 353 in the 2nd round), indicating frequent failures to provide valid answers. This issue is eliminated in both SFT and CC-SFT models. The original model's accuracy for class A decreases from 112 correct predictions in the 1st round to 109 in the 2nd round. In contrast, the SFT model

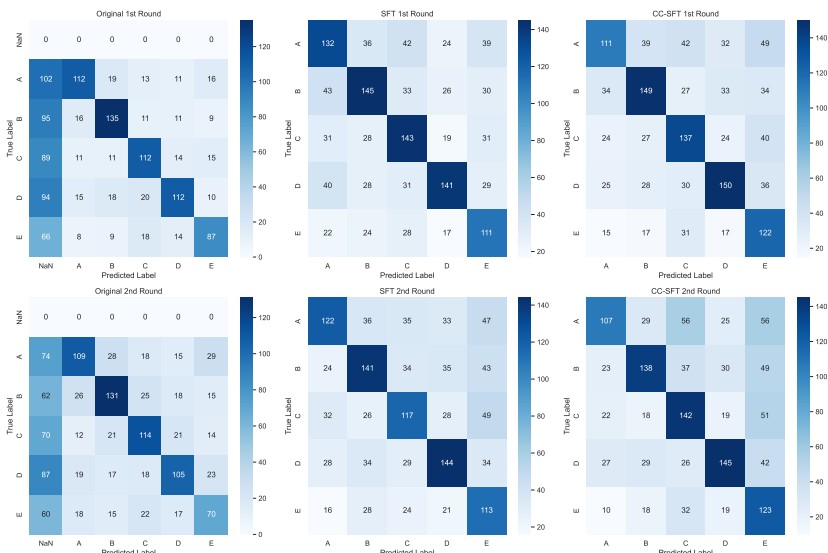

Figure 3: Comparison of 1st-round and 2nd-round confusion matrices generated by Llama v3.1 on MedQA-USMLE. *Nan* (not a number) indicates that the response does not match any of the predefined choices.

improves from 132 to 122, while CC-SFT maintains more consistent performance (111 to 107). For class C, the original model drops from 135 to 131 correct predictions, SFT declines from 143 to 117, but CC-SFT improves from 137 to 142. The CC-SFT model demonstrates the most stable performance across rounds, particularly for classes B (149 to 138) and D (150 to 145). Notably, CC-SFT reduces misclassifications in the second round compared to SFT. For example, misclassifications of true D as E decrease from 34 (SFT) to 26 (CC-SFT), and true E as C reduce from 24 (SFT) to 18 (CC-SFT). These results quantitatively demonstrate CC-SFT's effectiveness in maintaining consistent and accurate responses across multiple dialogue turns.

## 6 CONCLUSION

In this work, we address the critical issue of conversational inconsistency in LLMs by introducing a novel Conversationally Consistent Supervised Fine-Tuning method. Our approach, which explicitly accounts for two-turn conversations and incorporates a Wasserstein distance-based consistency loss, demonstrated significant improvements in maintaining coherent responses across dialogue turns. Through extensive experiments on OpenBookQA, GSM8K, and MedQA-USMLE datasets using Llama v3.1, Mistral AI, and Gemma, we show that CC-SFT consistently outperforms standard fine-tuning, reducing flipping rates and enhancing second-round response accuracy. We provide theoretical convergence guarantees and analyze the impact of the consistency loss coefficient. Our work contributes to enhancing the reliability and trustworthiness of LLMs in multi-turn dialogues, particularly crucial for high-stakes applications in healthcare and legal domains. Future research directions include extending the method to longer conversation histories, exploring its applicability to other language model architectures, and investigating its impact on specific downstream tasks. By mitigating conversational inconsistency, this study paves the way for more dependable AI-assisted services and decision-support systems, bringing us closer to the goal of truly reliable and coherent conversational AI.

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

# A APPENDIX

**Theorem 1** (Convergence of Stochastic Gradient Descent for LLM Consistency Loss). *Let $\theta^*$ be the optimal parameter that minimizes $\mathcal{L}(\theta)$. Consider the stochastic gradient descent update rule: $\theta_{t+1} = \theta_t - \eta_t \nabla \mathcal{L}_t(\theta_t)$ where $\eta_t = \frac{\beta}{t+\gamma}$ is the learning rate at step t, with $\beta > \frac{1}{2\mu}$ and $\gamma = \max\{4L\beta, 1\}$. Then, for T iterations, we have:*

$$\mathbb{E}[\|\theta_T - \theta^*\|^2] \leq \frac{C}{T}$$

*where C is a constant depending on L, $\mu$, $\sigma$, $\beta$, and $\|\theta_0 - \theta^*\|$.*

*Proof.* Let's proceed step by step:

1) Define the error at step $t$ as $e_t = \theta_t - \theta^*$. We want to bound $\mathbb{E}[\|e_T\|^2]$.

2) Using the update rule, we can write:

$$e_{t+1} = e_t - \eta_t \nabla \mathcal{L}_t(\theta_t)$$

3) Taking the squared norm of both sides:

$$\|e_{t+1}\|^2 = \|e_t\|^2 - 2\eta_t \langle e_t, \nabla \mathcal{L}_t(\theta_t) \rangle + \eta_t^2 \|\nabla \mathcal{L}_t(\theta_t)\|^2$$

4) Taking expectations and using Assumption 3:

$$\mathbb{E}[\|e_{t+1}\|^2] = \mathbb{E}[\|e_t\|^2] - 2\eta_t \mathbb{E}[\langle e_t, \nabla \mathcal{L}(\theta_t) \rangle] + \eta_t^2 \mathbb{E}[\|\nabla \mathcal{L}_t(\theta_t)\|^2]$$

5) Using Lemma 1 and the fact that $\nabla \mathcal{L}(\theta^*) = 0$:

$$\langle e_t, \nabla \mathcal{L}(\theta_t) \rangle \geq \frac{\mu L}{\mu + L} \|e_t\|^2 + \frac{1}{\mu + L} \|\nabla \mathcal{L}(\theta_t)\|^2$$

6) Substituting this into the inequality from step 4):

$$\mathbb{E}[\|e_{t+1}\|^2] \leq (1 - 2\eta_t \frac{\mu L}{\mu + L})\mathbb{E}[\|e_t\|^2] - 2\eta_t (\frac{1}{\mu + L} - \frac{\eta_t}{2})\mathbb{E}[\|\nabla \mathcal{L}(\theta_t)\|^2] + \eta_t^2 \sigma^2$$

7) Choose $\eta_t = \frac{\beta}{t + \gamma}$ with $\beta > \frac{1}{2\mu}$ and $\gamma = \max\{4L\beta, 1\}$. This ensures $\frac{1}{\mu+L} - \frac{\eta_t}{2} > 0$ for all $t$.

8) Define $v_t = (t + \gamma)\mathbb{E}[\|e_t\|^2]$. We can show by induction that:

$$v_t \leq v_0 + \frac{C_1}{\beta \mu} \sum_{i=1}^{t} \frac{1}{i + \gamma - 1}$$

where $C_1$ is a constant depending on $L$, $\mu$, $\sigma$, and $\beta$.

9) Using the bound on the harmonic series:

$$\sum_{i=1}^{t} \frac{1}{i + \gamma - 1} \leq \log(t + \gamma) - \log(\gamma) + 1$$

10) Substituting this back into the inequality for $v_t$:

$$v_t \leq v_0 + \frac{C_1}{\beta \mu}(\log(t + \gamma) - \log(\gamma) + 1)$$

11) Finally, we can conclude:

$$\mathbb{E}[\|e_T\|^2] = \frac{v_T}{T + \gamma} \leq \frac{v_0 + \frac{C_1}{\beta\mu}(\log(T + \gamma) - \log(\gamma) + 1)}{T + \gamma} \leq \frac{C}{T}$$

where $C$ is a constant depending on $L$, $\mu$, $\sigma$, $\beta$, and $\|\theta_0 - \theta^*\|$.

This completes the proof. $\square$

**Theorem 2.** *The convergence rate of $\mathcal{O}(\frac{1}{T})$ for the expected squared error implies that the loss function $\mathcal{L}(\theta)$ converges to its minimum value at a rate of $\mathcal{O}(\frac{1}{\sqrt{T}})$.*

*Proof.* Let's proceed step by step to prove this corollary:

1) Recall that $\theta^*$ is the optimal parameter that minimizes $\mathcal{L}(\theta)$, and $\theta_T$ is the parameter after $T$ iterations of stochastic gradient descent.

2) From the main theorem, we have:

$$\mathbb{E}[\|\theta_T - \theta^*\|^2] \leq \frac{C}{T}$$

3) Since $\mathcal{L}(\theta)$ is $\mu$-strongly convex (from Assumption 1), we can use the property of strong convexity:

$$\mathcal{L}(\theta) - \mathcal{L}(\theta^*) \geq \frac{\mu}{2}\|\theta - \theta^*\|^2$$

4) Applying this inequality to our case:

$$\mathcal{L}(\theta_T) - \mathcal{L}(\theta^*) \geq \frac{\mu}{2}\|\theta_T - \theta^*\|^2$$

5) Taking expectations of both sides:

$$\mathbb{E}[\mathcal{L}(\theta_T) - \mathcal{L}(\theta^*)] \geq \frac{\mu}{2}\mathbb{E}[\|\theta_T - \theta^*\|^2]$$

6) Using the result from step 2:

$$\mathbb{E}[\mathcal{L}(\theta_T) - \mathcal{L}(\theta^*)] \geq \frac{\mu}{2} \cdot \frac{C}{T} = \frac{\mu C}{2T}$$

7) Now, let's use the $L$-Lipschitz continuity of the gradient (from Assumption 2). For Lipschitz continuous functions, we have:

$$\mathcal{L}(\theta) - \mathcal{L}(\theta^*) \leq \frac{L}{2}\|\theta - \theta^*\|^2$$

8) Applying this to our case and taking expectations:

$$\mathbb{E}[\mathcal{L}(\theta_T) - \mathcal{L}(\theta^*)] \leq \frac{L}{2}\mathbb{E}[\|\theta_T - \theta^*\|^2] \leq \frac{L}{2} \cdot \frac{C}{T} = \frac{LC}{2T}$$

9) Combining the results from steps 6 and 8, we have:

$$\frac{\mu C}{2T} \leq \mathbb{E}[\mathcal{L}(\theta_T) - \mathcal{L}(\theta^*)] \leq \frac{LC}{2T}$$

10) This shows that $\mathbb{E}[\mathcal{L}(\theta_T) - \mathcal{L}(\theta^*)] = \mathcal{O}(\frac{1}{T})$

11) To get from $\mathcal{O}(\frac{1}{T})$ to $\mathcal{O}(\frac{1}{\sqrt{T}})$, we can use Jensen's inequality, which states that for a concave function $f$:

$$f(\mathbb{E}[X]) \geq \mathbb{E}[f(X)]$$

12) The square root function is concave, so we can apply Jensen's inequality:

$$\sqrt{\mathbb{E}[\mathcal{L}(\theta_T) - \mathcal{L}(\theta^*)]} \geq \mathbb{E}[\sqrt{\mathcal{L}(\theta_T) - \mathcal{L}(\theta^*)}]$$

13) From step 10, we know that $\mathbb{E}[\mathcal{L}(\theta_T) - \mathcal{L}(\theta^*)] = \mathcal{O}(\frac{1}{T})$. Therefore:

$$\sqrt{\mathbb{E}[\mathcal{L}(\theta_T) - \mathcal{L}(\theta^*)]} = \mathcal{O}(\frac{1}{\sqrt{T}})$$

14) Combining this with the result from step 12:

$$\mathbb{E}[\sqrt{\mathcal{L}(\theta_T) - \mathcal{L}(\theta^*)}] = \mathcal{O}(\frac{1}{\sqrt{T}})$$

Therefore, we have shown that the loss function $\mathcal{L}(\theta)$ converges to its minimum value at a rate of $\mathcal{O}(\frac{1}{\sqrt{T}})$. $\square$

