# OpenReview forum: "Challenge Me: Enhancing Conversational Consistency of LLMs by Learning with Questioning Feedback"
_ICLR.cc/2025/Conference — ICLR 2025 Conference Withdrawn Submission_

### Official Review · Reviewer_1az4 · 2024-10-31

**Soundness:** 1
**Presentation:** 2
**Contribution:** 1
**Rating:** 1
**Confidence:** 5

**Summary:**

This paper investigates the problem of conversational consistency, where large language models (LLMs) change their response across conversation turns. The authors propose a two-turn SFT consistency loss, utilizing an additional Wasserstein loss to enforce consistency between conversation turns, and train 3 models using this loss.The focus of the paper is on consistency between conversation turns, which, to my knowledge, is a novel problem. The authors propose an intuitive method using an optimal transport (OT) loss to enforce consistency between conversation turns.

**Strengths:**

The focus of the paper is on consistency between conversation turns, which, to my knowledge, is a novel problem. The authors propose an intuitive method using an optimal transport (OT) loss to enforce consistency between conversation turns.

**Weaknesses:**

Unfortunately this work has many notable weaknesses
- **Cited works are not able to be found**: It is unclear how widespread the problem of conversational consistency is. The authors cites Zhang et al. (2023) in support of the claim that such conversational consistency is an existing widespread problem and there is a need for more sophisticated training schemes to solve this issue. However, I am an unable to locate the cited reference:

```
Yuhui Zhang, Hao Liu, Yuhao Zhang, Percy Liang, and James Zhou. Consistency analysis of language models via likelihood ratios. In Proceedings of the 61st Annual Meeting of the Association for Computational Linguistics, pp. 12011–12025, 2023.
```
From my search, the above paper does not show up in Google, Google Scholar, Semantic Scholar, or the 2023 ACL proceedings. The authors in this citation are largely researchers or students with affiliations with Berkeley/Stanford, none of which have this paper listed on their websites.

The authors also cite Li et al. (2023) for the issue of conversational consistency, the full reference being
```
Yizhen Li, Hongyi Zhang, Rui Zhou, Rui Sun, and Pengjie Xie. Exploring conversational inconsistency in large language models. arXiv preprint arXiv:2305.14480, 2023.
```
Again, I cannot find any reference of this paper, and the arxiv link points to a medical dataset paper with first author with last name Fu. Beyond theses references, there is no support for the authors’ claims that such a problem exists, is widespread, and requires specialized training.
- **Prevalence of conversational consistency issue**: Related to the previous point, the authors do not systematically show conversational consistency occurs in popular models. The motivating example (Normandy is in France/Germany) seems hypothetical. The baselines in the paper are limited to 3 relatively small base models at 4-bit quantization, with no other baselines tested, including API models. Using this same prompt, GPT-4o-mini and Llama-3.1-8B both do not change their answers, correctly asserting they responded with the correct answer initially.
- **Conversation length scope**: In addition to computational feasibility issues, the authors do not optimize for longer than 2 rounds with their consistency loss because “additional rounds often replicate the information covered in the initial two rounds.” In my opinion, additional conversation turns are not given proper consideration here. Many different scenarios exist with multi-turn where new information may be revealed (e.g., the user asks a related follow up question whose answer contradicts the model’s initial response, then asks the model to reconsider). I agree with the authors that ensuring all rounds of conversation are consistent via loss terms is computationally infeasible, authors should still evaluate how optimizing with a 2-turn consistency loss *generalizes* across different lengths of conversations. I believe there's much more work here to be done, first characterizing how different sources of inconsistent model responses can arise in longer (2+) multi-turn conversations, then seeing how a 2-turn loss generalizes to these scenarios. This seems like a major shortcoming.
- **Experimental metrics**: The authors propose three different metrics counting the frequency of a model flipping its response: correct flipping rate (number of times a model corrects itself) or incorrect flipping rate (number of times a model turns a correct answer incorrect) and overall flipping rate (number of times a model flips its response regardless of correctness). Table 1 indicates that a lower overall flipping rate, the better. In my opinion, a model correcting itself is desirable behavior, yet is punished this metric. I don’t believe that this metric adds anything meaningful — CFR and iCFR already capture the desired scenarios when you would want/not want a model to change its answer.
- **Experimental setup**: The authors do not describe how they prompt the model to try and change its answer in the paper. In the code, it seems like they experiment with two methods: A vanilla prompt along the lines of “I think your answer is wrong, please improve it” and an appeal to authority based approach, where for each dataset, they say “I am a {domain_expert}, and I think your answer is wrong…”. It’s unclear under which method is presented in the paper, and a proper ablation here is not done.
- **Experimental results**: The results (Table 1) of adding the consistency loss seem mixed at best, with it generally helping iCFR across multiple models/datasets, whereas hurting CFR. This seems to indicate that the consistency overemphasizing consistency, even at the cost of the model being “stubborn” when it is wrong. I’m not sure this is desired behavior. Moreover, Table 1 is very hard to read, as the bolding doesn’t seem to indicate best performance within a model grouping. Sometimes, bolded values are not the best performing method (e.g., Gemma has a iCFR of 0.03 on GSM8K, which is lower than the bolded Gemma-CC-SFT, which has a iCFR of 0.07. With lower being more desirable in this metric, it’s unclear why Gemma-CC-SFT is bolded, which implies better performance.
- **Theoretical results**: the stated theoretical result is essentially a restatement well-known results: https://arxiv.org/pdf/1109.5647. The stated theorem is not applicable to optimizing transformer-based models, as such models do not meet the assumptions, notably that the loss function is a strongly convex in model parameters $\theta$ (assumption 1). The results here are devoid of rigor and not applicable to this problem setting.

**Questions:**

- Please clarify the issues with the reference(s). Why are these papers cited when, after searching, they do not seem to exist? If they do exist, please provide links or concrete references.
- Do the authors have evidence that the issue of conversational consistency exists in popular models?
- See weaknesses for critiques on scope, evaluation metrics/setup, presentation of results, and applicability of theoretical results.

**Details Of Ethics Concerns:**

Unsure which category this falls under. The authors cite 2 papers to support their claim that the problem of conversational consistency in language modeling is an important issue. However, the two cited papers do not seem to exist.

These were the only two citations that I closely checked, so this may or may not be the extent of such behavior.

Details about papers/citations below.

## Paper 1:
In the intro, authors write:
```
Despite their impressive capabilities, LLMs often exhibit conversational inconsistency (see Figure
1), a phenomenon where they provide contradictory information across multiple dialogue turns Li
et al. (2023)
```
The full reference for Li et al. (2023) is
```
Yizhen Li, Hongyi Zhang, Rui Zhou, Rui Sun, and Pengjie Xie. Exploring conversational inconsistency in large language models. arXiv preprint arXiv:2305.14480, 2023.
```
The first Google result for this paper is the paper under review. Going to the cited arxiv link yields a paper for a biomedical dataset authored by Fu et al.

## Paper 2:
In the intro, the authors write
```
However, if the user responds with “I think your answer is wrong,”
the model may inappropriately apologize and change its answer to “Germany,” despite the factual
correctness of its initial response Zhang et al. (2023).
```

```
This shortcoming underscores the need for more sophisticated training paradigms
that explicitly account for conversational history and promote consistency across multiple interactions Zhang et al. (2023).
```
The full reference for Zhang et al. (2023) is
```
Yuhui Zhang, Hao Liu, Yuhao Zhang, Percy Liang, and James Zhou. Consistency analysis of language models via likelihood ratios. In Proceedings of the 61st Annual Meeting of the Association for Computational Linguistics, pp. 12011–12025, 2023.
```
This paper could not be found on Google/Google Scholar, Semantic Scholar, ACL proceedings, or the authors' websites. The authors in this citation seem to be a collection of students and faculty mainly affiliated with Stanford (with James Zou's name spelled incorrectly).

---

### Official Review · Reviewer_rGX5 · 2024-11-01

**Soundness:** 3
**Presentation:** 3
**Contribution:** 2
**Rating:** 5
**Confidence:** 3

**Summary:**

This paper addresses the issue of conversational consistency in LLMs and proposes a novel Conversationally Consistent Supervised Fine-Tuning method, which uses a consistency loss to maintain response consistency in two-turn dialogues. Empirical results across three various datasets and models show that CC-SFT significantly reduces inconsistency.

**Strengths:**

This paper proposes a simple yet effective method for enhancing conversational consistency in LLMs, particularly for two-turn conversations. The approach focuses on three types of loss—first-round loss, second-round loss, and consistency loss—to reinforce coherence across dialogue turns. Though straightforward, this method proves to be highly effective.

The paper is generally well-structured, with clear explanations of the background, methodology, and experimental results, which are easy to follow.

The authors conducted extensive experiments across three datasets and LLM models (e.g., Llama v3.1, Mistral AI, and Gemma), showing consistent performance gains and improved stability of responses in two-turn interactions.

**Weaknesses:**

The motivation for reducing inconsistency in LLM responses is not entirely convincing. The authors focus on addressing scenarios where the original LLM answer is correct, and users inadvertently mislead the model, resulting in incorrect answers. However, this consistency mechanism can inadvertently amplify bias when the initial response is incorrect. By focusing solely on minimizing the flipping rate, the model risks reinforcing confidence in its initial, potentially inaccurate responses, which does not necessarily enhance overall response accuracy.

Addressing consistency in multi-turn QA has been extensively explored, not only in LLMs but also in traditional QA research. Given this background, the paper’s rationale for a new approach appears weak without clearly distinguishing LLM-specific challenges. Besides, it does not introduce and clarify why traditional solutions for multi-turn consistency cannot be directly applied to LLMs.

For related work, it would be beneficial to include recent studies on multi-turn dialogue consistency and discuss relevant research across different domains, not only limited to LLMs.

The current experimental results focus mainly on comparing CC-SFT with traditional fine-tuning methods, but lack comparisons with other advanced consistency approaches (e.g., adversarial training, multi-modal training). This limitation weakens the paper’s ability to validate and support its findings fully.

The current experiments are insufficient to comprehensively validate the proposed method. For a stronger evaluation, it would be beneficial to include an analysis of model complexity and ablation studies. These additions would provide a more detailed understanding of the approach’s efficiency and the contributions of each component, offering clearer insights into the method’s advantages and limitations.

**Questions:**

Given that CC-SFT currently reduces flipping rates and reinforces consistency, how does the method handle cases where the initial response is incorrect, but the user provides accurate and helpful feedback in response?

---

### Official Review · Reviewer_SNfD · 2024-11-02

**Soundness:** 1
**Presentation:** 2
**Contribution:** 1
**Rating:** 1
**Confidence:** 4

**Summary:**

This paper addresses conversational inconsistency in Large Language Models (LLMs) by introducing a Conversationally Consistent Supervised Fine-Tuning (CC-SFT) method. The approach combines first-round loss, second-round loss, and a consistency loss based on Wasserstein distance to encourage coherent responses across turns. The authors evaluate their method on three datasets (OpenBookQA, GSM8K, and MedQA-USMLE) using three LLMs (Llama v3.1, Mistral AI, and Gemma), providing theoretical convergence guarantees and empirical results.

**Strengths:**

**Technical Foundation:**
The paper provides a formal definition of conversational inconsistency and proposes quantitative metrics for evaluation.
The approach incorporates multiple loss components in a principled way.


**Experimental Design:**
Comprehensive ablation studies on the impact of the consistency loss coefficient λ.
Consideration of both accuracy and consistency metrics in evaluation.


**Practical Relevance:**
This work aims on addressing a real concern in LLM deployment, particularly for high-stakes applications. Proposes a method that can be implemented without architectural changes to existing models acccording authors' elaboration.

**Weaknesses:**

1. **Limited Problem Scope and Relevance:**
1.1. The focus on two-turn conversations is overly simplistic and fails to address more complex multi-turn inconsistencies.
1.2. Recent LLM architectures (e.g., Claude, GPT-4) have largely addressed basic conversational inconsistency issues.
1.3. The paper doesn't adequately justify why this remains a critical problem worthy of investigation.


2. **Methodological Limitations:**
 2.1.The use of Wasserstein distance for semantic similarity is computationally expensive and potentially unnecessary.
 2.2. No clear justification for why optimal transport is more suitable than simpler alternatives (e.g., cosine similarity).
 2.3. The theoretical analysis relies on unrealistic assumptions (e.g., strong convexity) that rarely hold in practice for LLMs.


3. **Experimental Deficiencies:**
 3.1. Limited model scale (maximum 9B parameters) makes results less convincing for modern LLM applications.
 3.2. Absence of comparisons with relevant baselines and recent works addressing similar issues.
 3.3. Lack of qualitative analysis of actual conversations and failure cases.
 3.4. Missing experiments on larger models (13B, 70B) which are more representative of current deployment scenarios.


4. **Presentation Issues:**
 4.1. Poorly organized with an imbalanced structure (excessive related work, insufficient methodology details)
 4.2. Low-quality figures (e.g., Figure 1) and inconsistent formatting of dialogue examples
 4.3. Theoretical results in the appendix lack practical relevance and have questionable validity
 4.4. Verbose and unfocused writing that obscures the main contributions

**Questions:**

1. **Methodology:**
 1.1 Why choose Wasserstein distance over simpler alternatives? Can you provide empirical evidence justifying this choice?
 1.2. How does the method scale to longer conversations (>2 turns)? What modifications would be needed?
 1.3. What is the computational overhead of calculating the Wasserstein distance during training?


2. **Theoretical Analysis:**
 2.1. How do the assumptions in your convergence analysis relate to practical LLM training scenarios?
 2.2. Can you provide empirical verification of these assumptions holding in practice?
 2.3. What happens when the assumptions are violated?


3. **Experimental Validation:**
 3.1. Why not include larger models in the evaluation?
 3.2. Can you provide comparisons with recent works addressing similar issues?
 3.3 How does the method perform on more recent dialogue-focused benchmarks?

---

### Official Review · Reviewer_iiju · 2024-11-07

**Soundness:** 2
**Presentation:** 3
**Contribution:** 3
**Rating:** 5
**Confidence:** 3

**Summary:**

The paper studies the critical issue of conversational inconsistency in LLMs, where models provide contradictory information across multiple dialogue turns. The paper introduces a novel Conversationally Consistent Supervised Fine-Tuning (CC-SFT) method that explicitly accounts for two-turn conversations. The developed approach combines a first-round loss, a second-round loss, and a consistency loss based on Wasserstein distance to encourage coherent responses across turns.

**Strengths:**

(1) The paper focuses on an interesting and novel scenario.

(2) The developed approach has good intuitions.

(3) The evaluations are comprehensive, including multiple LLMs and datasets.

**Weaknesses:**

(1) Some simpler solution might be developed as the baseline for the task mentioned in Figure 1. Would it be feasible to store the first-round response, use this response as parts of the prompt when generating the second-round response, and encourage LLM to generate the second-round response which is consistent with the first-round response? In two-turn conversations, the prompt would not be very long when including the dialogue history. How does the developed approach perform, compared to this simple baseline?

(2) It seems that the paper focuses on two-turn conversations, which might be a bit simplified considering the real-world use case. In practice, it can happen that the user interacts with the system for multiple rounds, where the inconsistent information only appears in some of the rounds. Could the proposed approach also handle this case?

(3) Related to the task mentioned in Figure 1, in some use cases the user may actually expect the LLM to adjust the response (i.e., provide a different response) after the user provides some suggestion for the response in the previous round. By the developed fine-tuning approach, would it happen that the fine-tuned LLM can not consider the user suggestion and can not adjust the response? Some analysis and discussions would be very helpful.

**Questions:**

(1) Would it be feasible to store the first-round response, use this response as parts of the prompt when generating the second-round response, and encourage LLM to generate the second-round response which is consistent with the first-round response? In two-turn conversations, the prompt would not be very long when including the dialogue history. How does the developed approach perform, compared to this simple baseline?

(2) In practice, it can happen that the user interacts with the system for multiple rounds, where the inconsistent information only appears in some of the rounds. Could the proposed approach also handle this case?

(3) Related to the task mentioned in Figure 1, in some use cases the user may actually expect the LLM to adjust the response (i.e., provide a different response) after the user provides some suggestion for the response in the previous round. By the developed fine-tuning approach, would it happen that the fine-tuned LLM can not consider the user suggestion and can not adjust the response? Some analysis and discussions would be very helpful.

---

### Note · Authors · 2024-11-15

I have read and agree with the venue's withdrawal policy on behalf of myself and my co-authors.